SciPost Physics

# No superconductivity in Pb$_9$Cu$_1$(PO$_4$)$_6$O found in orbital and spin fluctuation exchange calculations

Niklas Witt[1,2†] ⓘ, Liang Si[3,4] ⓘ, Jan M. Tomczak[5,4] ⓘ, Karsten Held[4*] ⓘ
and Tim O. Wehling[1,2‡*] ⓘ

**1** I. Institute for Theoretical Physics, University of Hamburg, Notkestraße 9-11, 22607 Hamburg, Germany
**2** The Hamburg Centre for Ultrafast Imaging, Luruper Chaussee 149, 22607 Hamburg, Germany
**3** School of Physics, Northwest University, Xi'an 710127, China
**4** Institute of Solid State Physics, TU Wien, 1040 Vienna, Austria
**5** Department of Physics, King's College London, Strand, London WC2R 2LS, United Kingdom
† niklas.witt@physik.uni-hamburg.de
‡ tim.wehling@physik.uni-hamburg.de
* These authors contributed equally.

September 5, 2023

## Abstract

Finding a material that turns superconducting under ambient conditions has been the goal of over a century of research, and recently Pb$_{10-x}$Cu$_x$(PO$_4$)$_6$O aka LK-99 has been put forward as a possible contestant. In this work, we study the possibility of electronically driven superconductivity in LK-99 also allowing for electron or hole doping. We use an *ab initio* derived two-band model of the Cu $e_g$ orbitals for which we determine interaction values from the constrained random phase approximation (cRPA). For this two-band model we perform calculations in the fluctuation exchange (FLEX) approach to assess the strength of orbital and spin fluctuations. We scan over a broad range of parameters and enforce no magnetic or orbital symmetry breaking. Even under optimized conditions for superconductivity, spin and orbital fluctuations turn out to be too weak for superconductivity anywhere near to room-temperature. We contrast this finding to non-self-consistent RPA, where it is possible to induce spin singlet $d$-wave superconductivity at $T_c \geq 300$ K if the system is put close enough to a magnetic instability.

## 1 Introduction

The recent papers by Lee *et al.* [1, 2] reporting that Pb$_{10-x}$Cu$_x$(PO$_4$)$_6$O with $0.9 < x < 1.1$ (aka LK-99) is a room-temperature superconductor at ambient-pressure have been followed by extraordinary experimental and theoretical efforts. It even caught the attention of major news outlets and went viral on social media.

Experimental efforts to reproduce these measurements have led to mixed results. Some experiments confirm a jump in the conductivity as in the original work [1–3], albeit at a different temperature [4] or at a similar temperature but with an insulating resistivity at lower temperatures [5]. Also the levitation of Lee, Kim, *et al.* [2] or their diamagnetic response has been reproduced by other groups [6–8].

In stark contrast, other experiments find an insulator [4, 9, 10] and a paramagnetic behavior [9, 10]. Most experiments show a gray-black color, but a recent one reported transparency [11]. Matters become even more complicated, since –to the best of our knowledge–

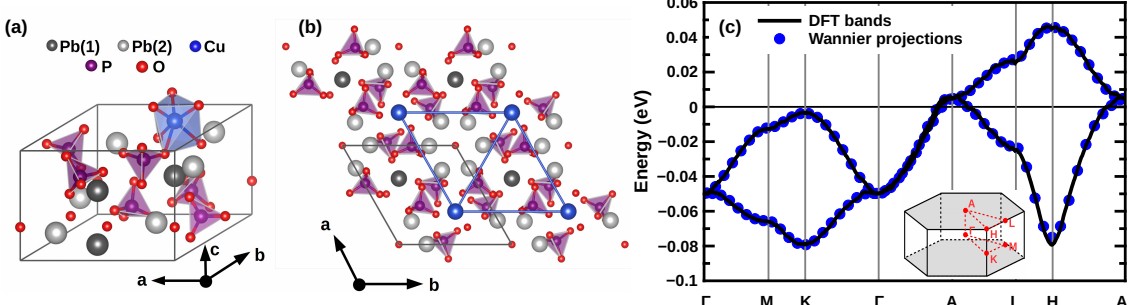

Figure 1: **Crystal structure and electronic bands of LK-99.** (a) DFT-relaxed structure of $Pb_9Cu(PO_4)_6O$; (b) View along $z$-direction for the $2\times2\times1$ supercell of (a); (c) DFT and Wannier band structures, the inset shows the $k$-path selected for band plotting. The DFT and Wannier projection data are adopted from Ref. [13].

hitherto no single phase sample has been synthesized, as evidenced by x-ray diffraction (XRD) [2, 4, 10] — not to speak of a single crystal. How can one make sense of these seemingly contradictory results?

Based on the observation that the Coulomb interaction $U$ dominates over the kinetic energy or bandwidth $W$, with $U/W$ of $\mathcal{O}(10)$, two of us [12] concluded that LK-99 must be a Mott or charge transfer insulator irrespective of $x$. This has been confirmed in independent calculations [13–15] using density functional theory (DFT) in combination with dynamical mean-field theory (DMFT) [16–19]. Likewise, DFT+$U$ calculations [13, 20–23] show a splitting into Hubbard bands. However, here a magnetic symmetry breaking (ordering) and a crystal structure or distortion that lifts the degeneracy of the two Cu $e_g$ orbitals crossing the Fermi energy is required. All of this confirms: Pure LK-99 is a Mott or charge transfer insulator. Thus, simultaneous to experimental efforts, theoretical simulations explain the insulating and paramagnetic behavior.

At the same time, the metallic (and potentially also the superconducting) behavior could be explained if LK-99 is electron or hole-doped, e.g., with $y > 0$ and $z \neq 0$ in $Pb_{10-x}Cu_x(P_{1-y}S_yO_4)_6O_{1+z}$. At least metallic behavior and a gray-black or similar color [2, 4, 10] are then to be expected. A noteworthy other explanation has been put forward by Zhu *et al.* [24] and Jain [25]: the resistivity jump and $\lambda$-like feature in the specific heat could be simply caused by $Cu_2S$ which is clearly present as a secondary phase in XRD measurements [2, 4, 10, 24].

The most important question however remains open: Is electron or hole doped LK-99 superconducting? Here, experiment is inconclusive and calculations have so-far been very limited: Enforcing superconductivity with a Bardeen, Cooper and Schrieffer(BCS) [26] Hamiltonian, Tavakol *et al.* [27] find $f$-wave pairing. Oh and Zhang [28] obtain a self-consistently determined $s$-wave pairing in an effective $t-J$ model at zero temperature.

In this paper, we aim at giving a more definite answer regarding superconductivity. Based on an *ab initio* derived two-band model [13][1], which suffices for a Mott insulator, we employ the fluctuation exchange (FLEX) [30–32] approach. Our results are very clear: we can exclude superconductivity — at least superconductivity based on orbital and spin fluctuations in the two-band model of LK-99.

---

[1]A first tight-binding parametrization mimicking the DFT dispersion was derived in Ref. [29] purely from symmetry considerations

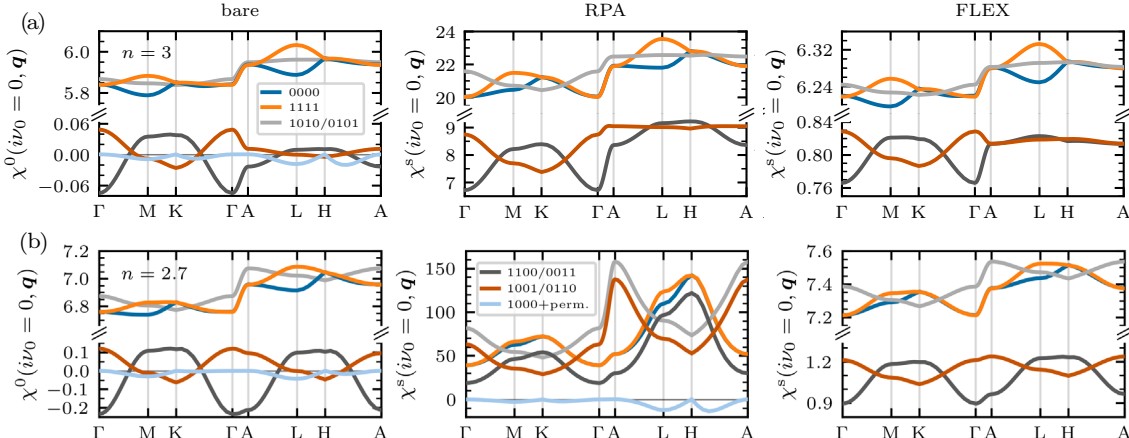

Figure 2: **Susceptibilities and spin fluctuations.** Plots of the static non-interacting susceptibility $\chi^0$ (left column), RPA (middle column) and FLEX spin susceptibility $\chi^s$ (right column) components as function of momentum $\boldsymbol{q}$ obtained at temperature $T = 300\,\mathrm{K}$ for (a) the nominal filling of LK-99 ($n = 3$) and (b) in a hole-doped case ($n = 2.7$). The RPA and FLEX calculations assumed the ratio $U/J = 0.183$ as in cRPA, but the interaction magnitude tuned to $U = 0.115\,\mathrm{eV}$, which is in RPA near the magnetic instability and thus in the vicinity of the point of maximal spin fluctuations. Note the very different scales of the spin susceptibilities in RPA and FLEX.

## 2 Results

For the structure of LK-99 shown in Fig. 1(a,b), where the Cu atoms form a triangular sublattice, DFT calculations [12, 21, 33, 34] yield the low-energy *ab-initio* band structure shown in Fig. 1(c). The two lowest-energy bands are well captured by the Wannier functions projected onto the $d_{yz}$ and $d_{xz}$ orbitals in the energy window of [-0.1, 0.1] eV. These bands disperse over a bandwidth of the order of ∼120 meV. Calculations of the Coulomb interaction tensor in the constrained random phase approximation (cRPA) yield local intra-orbital Hubbard repulsion $U_{\mathrm{cRPA}} = 1.8\,\mathrm{eV}$, an inter-orbital $U'_{\mathrm{cRPA}} = 1.14\,\mathrm{eV}$, and a Hund's exchange $J_{\mathrm{cRPA}} = 0.33\,\mathrm{eV}$ (c.f. section A.2). These interactions exceed the electronic bandwidth by far and put LK-99 into the regime of strong electron correlations — in line with recent DMFT [13–15], and DFT+$U$ studies [13, 20–23].

Here, our goal is to establish an *upper boundary* for spin- and orbital-fluctuation-driven superconductivity (SC). To this end, we resort to a RPA and FLEX analysis. Fig. 2 compares the static momentum (**q**)-dependent non-interacting susceptibility $\chi^0(\mathbf{q})$ (left column) to the spin susceptibility $\chi^s(\mathbf{q})$ as obtained from RPA and FLEX for an interaction with an $U/J$ ratio as in cRPA. First, we study the system for a scaled-down overall magnitude of the interaction matrix elements with $U = 0.115$eV. This regime is close to the RPA's magnetic instability, where we expect the strongest tendencies towards spin-fluctuation-driven superconductivity. We consider two different dopings: the nominal filling of $n = 3$ electrons per unit cell (upper panel) an hole doping to $n = 2.7$. At both doping levels, the RPA spin susceptibility exceeds the non-interacting susceptibility by factors of 4 to 20, respectively, signaling a correspondingly strong Stoner enhancement. Indeed, this strong Stoner enhancement confirms that our scaled down interaction ($U = 0.115$ eV) is close to the RPA Stoner instability and thus also in the vicinity of the region, where we potentially expect the strongest tendency for spin- or orbital-fluctuation-driven superconductivity.

Indeed, we find magnetic instabilities over a wide range of fillings $2 < n < 3.3$ in RPA that set in already for interactions of the order of the bandwidth, $U \approx 0.12\,\mathrm{eV} \ll U_{\mathrm{cRPA}}$. RPA

therefore puts LK-99 deeply into a magnetically ordered state.

However, care must be taken since renormalization effects and vertex corrections can decisively impact phase diagrams and could hypothetically suppress magnetic order. The FLEX method takes into account renormalization effects stemming from the scattering of electrons with spin, orbital and charge fluctuations in terms of a diagrammatic ladder resummation. The resultant FLEX spin susceptibilities (Fig. 2 right columns) are indeed much smaller than their RPA counterparts and show much weaker Stoner enhancement. A further increase of the Hubbard $U$ leads to a *reduced* spin susceptibility in FLEX related to a reduction of the electronic quasiparticle weight.

Scattering of electrons off spin and orbital fluctuations can mediate superconductivity [35, 36]. Within RPA and FLEX, the resultant pairing interactions in the singlet and triplet channel, cf. Eq. (10), are controlled by the spin and charge susceptibilities and grow as $\chi^{s,c}$ increase. A transition into a superconducting state is indicated by the leading eigenvalue, $\lambda_{SC}$, of the linearized Eliashberg equation, see Eq. (9) below, reaching unity. Fig. 3 shows $\lambda_{SC}$ as obtained from RPA for different doping levels and ratios $J/U$ [2] as a function of interaction strength $U$ at a temperature of $T = 300$ K. We see that essentially at any hole-doping level in the range of $2.1 < n < 3$ fine tuning of the interaction seemingly leads to a superconducting instability even at 300 K. The resultant dominant order parameter in RPA is visualized in Fig. 3(f). This order parameter is in the spin singlet channel and involves significant inter-orbital pairing as well as sign changes between different momentum or orbital components. Electron doping ($n = 3.3$), on the contrary, is detrimental to the formation of SC pairing even within RPA.

Analyzing the RPA results more closely, we find that the superconducting tendencies exclusively occur, when closely approaching a Stoner instability, where the effective interaction strength [see Eq. (10) below] becomes unphysically large.

Indeed, renormalization effects strongly limit the maximal strength of the effective pairing interaction: This effect manifests dramatically in the Eliashberg eigenvalues $\lambda_{SC}$ achievable in FLEX. Scanning a wide range of interaction strengths and dopings, we obtain the temperature dependent $\lambda_{SC}$ in FLEX shown in Fig. 3(c-e). Anywhere close to room temperature we have $\lambda_{SC}$ merely of the order of $10^{-2}$, which is far away from any superconducting instability. Even if we go down in temperature to 20 K, $\lambda_{SC}$ at best reaches values on the order of $10^{-1}$. If Coulomb interaction mediated spin- or orbital-fluctuation-driven superconductivity sets in, it could only do so at significantly reduced temperatures.

The stark contrast between non-self-consistent RPA, where it is possible to seemingly find SC states with $T_c \geq 300$ K, and FLEX where SC is absent anywhere close to room temperature shows that the loss of electronic coherence due to scattering between electrons and spin fluctuations, which is missing in RPA, is responsible for the absence of superconductivity in the two-band model of LK-99.

## 3   Discussion and Conclusion

Our FLEX simulations do not find any superconducting instability in an *ab initio* derived two-band description of LK-99 — despite of a large range of dopings and interactions being considered. How unlikely does this render spin- or orbital-fluctuation-driven superconductivity in LK-99?

It is clear that also FLEX is an approximate method with shortcomings in the regime of strong correlations, including the failure to describe Hubbard bands correctly. Importantly, however, previous comparisons of FLEX against DMFT-based treatments of strongly correlated

---

[2] The ratio $J/U = 0.183$ stems from our cRPA simulations; the interaction estimate by Si *et al.* [13] leads to $J/U = 0.22$, whereas the cRPA calculations by Yue *et al.* [15] fall in-between with $J/U = 0.207$.

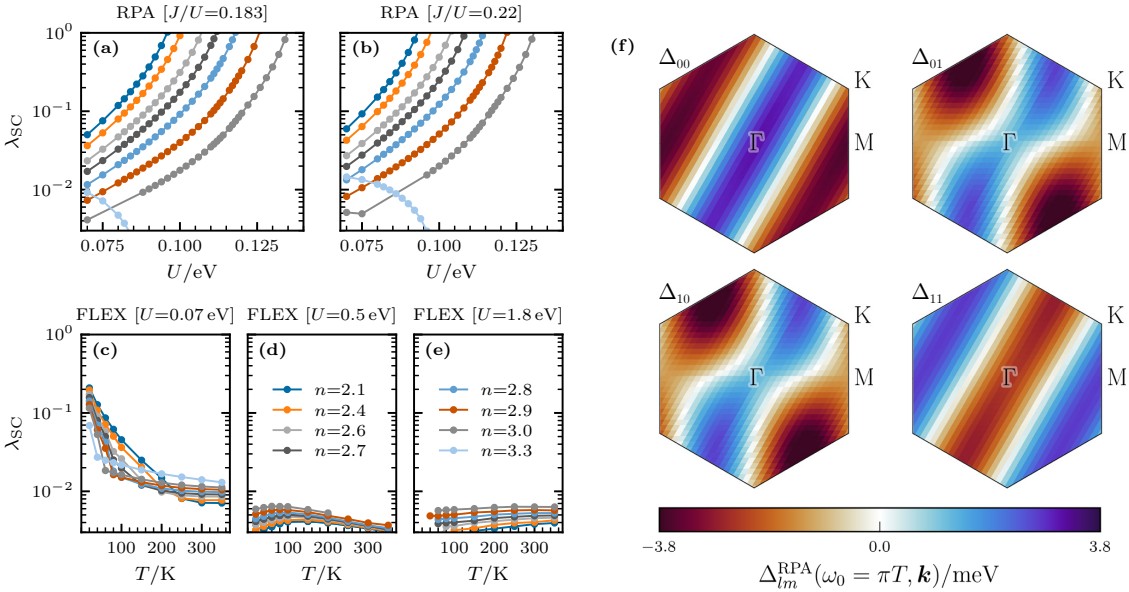

Figure 3: **Superconductivity and SC order parameters in RPA and FLEX.** Eigenvalues of the leading SC instability in linearized Eliashberg equation as obtained in (a,b) RPA and (c-e) FLEX for different dopings $n$. In RPA, $\lambda_{SC}$ is shown as function of interaction strength $U$ at temperature $T = 300\,\mathrm{K}$ for different ratios $J/U = 0.183$ (a) and 0.22 (b). Whenever a generalized Stoner instability is approached the SC eigenvalue reaches up to order $\lambda_{SC} \approx 1$ in RPA seemingly signaling an SC instability of the system. The associated gap function $\Delta$ is shown in panel (f), where the momentum dependence of each matrix element $\Delta_{lm}$ is shown in the $k_z = 0$ plane. FLEX calculations kept the ratio $J/U = 0.183$ fixed, and show $\lambda_{SC}$ as a function of temperature $T$ for different interaction strengths $U = 0.07\,\mathrm{eV}$ (c), $U = 0.5\,\mathrm{eV}$ (d), and $U = 1.8\,\mathrm{eV}$ (e). At room temperature we have $\lambda_{SC} \ll 10^{-1}$ ruling out any SC instabilities at this temperature.

superconductivity for the one-band Hubbard model [32] showed that achievable critical temperatures $T_c$ have the same magnitude in both approaches. Thus, within the two-band model considered here, room temperature superconductivity appears out of reach.

Still, two loopholes related to the two-band model by itself remain in principle open: First, the model considered here, assumes a periodic crystal with minimal unit cell comprising one idealized chemical composition $Pb_{10-x}Cu_x(P_{1-y}S_yO_4)_6O_{1+z}$ with $x = 1$, $y = 0$, $z = 0$ and optimized O and Cu positions [12]. This implies a triangular lattice of the Cu sites. However, several structures with different O and Cu positions are very close in energy [12] such that a disordered arrangement as suggested also by the XRD experiments [2, 4, 10] can be expected. While we cannot exclude disorder-enhanced superconductivity, it is not clear how a sufficient stiffness of the SC order parameter and a sufficient pairing strength to boost $T_c$ by more than an order of magnitude should be achievable here. Further, electron and hole doping of LK-99, corresponding to $y \neq 0$ and/or $z \neq 0$, is merely treated by changing the chemical potential in a rigid band approach.

Second, FLEX does not describe Hubbard bands. For a doped Mott insulator, however, we have quasiparticle renormalized Cu-$d$ bands crossing the Fermi energy. This situation and potentially arising superconducting instabilities *can* be described by FLEX. Qualitatively, we are thus on the safe side, since we committed ourselves to analyzing a broad range of parameters with largely different quasiparticle renormalizations. Still, a hole-doped charge transfer insulator, where the oxygen $p$ bands cross the Fermi energy and the lower copper $d$ Hubbard band

lies just below, cf. Ref. [13], cannot be described. At least for hole-doped cuprate superconductors, which are charge transfer insulators, O $p$ and Cu $d$ orbitals form a strongly hybridized single band—a situation [37, 38] which, then again, is in reach of FLEX. For electron doping, the Pb $p$ orbitals are too high in energy ($\geq 3$ eV) [12] for a charge-transfer arrangement with the upper Hubbard band.

Taken together, our study puts strong constraints on superconductivity in LK-99 and in particular excludes spin- an orbital-fluctuation-driven room temperature superconductivity in the two-band model of LK-99.

*Note added*—When completing this manuscript, a first single crystal of LK-99 has been synthesized and shows a non-magnetic insulating and transparent behavior [39], consistent with a Mott or charge transfer insulator [12–15].

# Acknowledgments

For the purpose of open access, the authors have applied a CC BY public copyright license to any Author Accepted Manuscript version arising from this submission.

**Author contributions**   N.W. performed RPA and FLEX calculations. L.S performed DFT calculations and the Wannierization. J.M.T. performed the cRPA calculations. N.W. and T.W. analyzed the RPA and FLEX results. N.W., K.H. and T.W. conceived the project. All authors discussed the results, physical implications and remaining caveats, and contributed to writing the manuscript.

**Funding information**   We, in particular, acknowledge funding via the Research Unit 'QUAST' by the Deutsche Foschungsgemeinschaft (DFG; project ID FOR5249, Project No. 449872909) and Austrian Science Fund (FWF, project ID I 5868). N. W. and T. W. further gratefully acknowledge funding by the Cluster of Excellence 'CUI: Advanced Imaging of Matter' of the DFG (EXC 2056, Project ID 390715994). K. H. has received additional funding through the FWF projects I 5398, P 36213, SFB Q-M&S (FWF project ID F86). L. S. is thankful for the starting funds from Northwest University and FWF project I-5398. J. T. acknowledges financial support through joint project I 6142 of FWF and the French National Research Agency (ANR). The RPA and FLEX calculations have been performed on the supercomputer Lise at NHR@ZIB as part of the NHR infrastructure; cRPA calculations have been done on the Vienna Scientific Cluster (VSC).

**Data availability**   The DFT and Wannierization data are available from the NOMAD repository XXX. The cRPA, RPA and FLEX data sets analyzed during the current study are available via XXX. Simulation data are available from the corresponding authors on request.

**Code availability**   VASP and Wien2K, used as the starting point of the relaxation and Wannierization, respectively, are commercial codes. The code for the Wannierization itself is publicly available at https://github.com/wien2wannier/wien2wannier. The computer code to perform FLEX calculations using the IR basis is publicly available under https://github.com/nikwitt/FLEX_IR.

**Competing interests**   The authors declare no competing interests.

# A  Methods

## A.1  Electronic structure

For a realistic simulation of putative superconducting properties in LK-99, we set up an effective low-energy theory via a Hamiltonian $\mathcal{H} = \mathcal{H}^0 + \mathcal{H}_{\text{int}}$. For the non-interacting part, $\mathcal{H}^0$, we use the *ab initio* derived two-orbital Wannier model of Si *et al.* [13]. It is given by

$$\mathcal{H}^0 = \sum_{i,j} \sum_{m,n} \sum_{\sigma} t_{im,jn} c_{im\sigma}^\dagger c_{jn\sigma}, \tag{1}$$

where $c_{im\sigma}^\dagger (c_{im\sigma})$ are the creation (annihilation) operators; and $i$, $j$ indicate unit cells, while $m$, $n$ are orbital indices, and $\sigma$ is the spin index. The full two-orbital hopping parameters $t_{im,jn}$ yield Fig. 1(c) and a truncated set of these is tabulated in Ref. [ [13]].

## A.2  Constrained Random Phase Approximation

We here compute the interacting part of the Hamiltonian from first principles. Specifically, we consider

$$\mathcal{H}_{\text{int}} = \frac{1}{4} \sum_{i} \sum_{\alpha_1 \alpha_2 \alpha_3 \alpha_4} \Gamma^0_{\alpha_1 \alpha_4, \alpha_3 \alpha_2} c_{i\alpha_1}^\dagger c_{i\alpha_2}^\dagger c_{i\alpha_3} c_{i\alpha_4} \tag{2}$$

where $i$ is a lattice site and the indices $\alpha_m = (\sigma, m)$ combine spin and orbital information. The bare vertex $\Gamma^0$ is expressed as

$$\begin{aligned}
\Gamma^0_{\alpha_1 \alpha_4, \alpha_3 \alpha_2} = &-\frac{1}{2} U^{\text{s}}_{m_1 m_4, m_3 m_2} \boldsymbol{\sigma}_{\sigma_1 \sigma_4} \cdot \boldsymbol{\sigma}_{\sigma_2 \sigma_3} \\
&+ \frac{1}{2} U^{\text{c}}_{m_1 m_4, m_3 m_2} \delta_{\sigma_1 \sigma_4} \delta_{\sigma_2 \sigma_3},
\end{aligned} \tag{3}$$

with the interaction matrices

$$U^{\text{s}}_{ll',nn'} = \begin{cases} U \\ U' \\ J \\ J \end{cases}, \quad U^{\text{c}}_{ll',nn'} = \begin{cases} U & (l = l' = n = n') \\ -U' + 2J & (l = n \neq n' = l') \\ 2U' - J & (l = l' \neq n' = n) \\ J & (l = n' \neq n = l') \end{cases}$$

in the spin (s) and charge (c) channel. The static matrix elements $U$, $U'$ and $J$ of the screened Coulomb interaction have then been computed with the constrained random phase approximation (cRPA) in the maximally localized Wannier basis [40], using $3 \times 3 \times 3$ reducible **k**-points, and including screening from orbitals up to $l = 3$ (2) for Pb,Cu (O,P). For the above two-orbital model, we find an intra-orbital Hubbard repulsion $U = 1.8$ eV, an inter-orbital $U' = 1.14$ eV, and a Hund's exchange $J = 0.33$ eV that are found to verify the symmetry relation $U' = U - 2J$. The reduction from the bare interactions $V = 12.7$ eV, $V' = 11.6$ eV, and $J_0 = 0.56$ eV, respectively, is larger than in the recent Refs. [11,15], possibly owing to our inclusion of more high-energy orbitals.

## A.3  Fluctuation exchange approach

To study the possibility of electronically-driven superconductivity, we employ the multi-orbital FLEX approximation [30,31]. FLEX is a conserving approximation that self-consistently incorporates spin and charge fluctuations by an infinite resummation of closed bubble and ladder

diagrams. Although FLEX cannot capture strong-coupling physics like the Mott-insulator transition, it works well in the presence of strong spin fluctuations.

We consider the multi-orbital formulation of FLEX without spin-orbit coupling [32, 41, 42] where we consider a local interaction Hamiltonian.

In the FLEX approximation, one solves the Dyson equation

$$\hat{G}(k)^{-1} = \left[ i\omega_n \mathbb{1} - (\hat{H}_0(\mathbf{k}) - \mu \mathbb{1}) \right]^{-1} - \hat{\Sigma}(k) \tag{4}$$

with the dressed Green function $G$, non-interacting Hamiltonian $H_0$, self-energy $\Sigma$, chemical potential $\mu$, and the four-momentum $k = (i\omega_n, \mathbf{k})$ containing crystal momentum $\mathbf{k}$ and Matsubara frequencies $\omega_n = (2n + 1)\pi k_\mathrm{B} T$. The hat denotes a matrix in orbital space, where $\mathbb{1}$ is the identity matrix. The interaction $V$ that enters the self-energy $\Sigma$ via

$$\Sigma_{lm}(k) = \frac{T}{N_\mathbf{k}} \sum_{q, l', m'} V_{ll', mm'}(q) G_{l'm'}(k - q) \tag{5}$$

consists of scattering off of spin and charge fluctuations given by

$$V = \frac{3}{2} \hat{U}^\mathrm{s} \left[ \hat{\chi}^\mathrm{s} - \frac{1}{2} \hat{\chi}^0 \right] \hat{U}^\mathrm{s} + \frac{1}{2} \hat{U}^\mathrm{c} \left[ \hat{\chi}^\mathrm{c} - \frac{1}{2} \hat{\chi}^0 \right] \hat{U}^\mathrm{c} \tag{6}$$

neglecting the constant Hartree-Fock term. The charge and spin susceptibility entering Eq. (5) are defined by

$$\hat{\chi}^{\mathrm{s,c}}(q) = \hat{\chi}^0(q) \left[ \mathbb{1} \mp \hat{U}^{\mathrm{s,c}} \hat{\chi}^0(q) \right]^{-1}, \tag{7}$$

with the irreducible susceptibility

$$\chi^0_{ll', mm'}(q) = -\frac{T}{N_\mathbf{k}} \sum_k G_{lm}(k + q) G_{m'l'}(k). \tag{8}$$

These equations are solved self-consistently with adjusting $\mu$ at every iteration to keep the electron filling fixed. We employ a linear mixing $G = \kappa G^\mathrm{new} + (1 - \kappa) G^\mathrm{old}$ with $\kappa = 0.2$ and defined self-consistency for a relative difference of $10^{-4}$ between the self-energy of two iteration steps. In all calculations, we used a $\mathbf{k}$-mesh resolution of $30 \times 30 \times 30$. For the imaginary-time and Matsubara frequency grids we applied the sparse-sampling approach [32, 43, 44] in combination with the intermediate representation (IR) basis [45–47], where we used an IR parameter of $\Lambda = 10^4$ and a basis cutoff of $\delta_\mathrm{IR} = 10^{-15}$.

To study the superconducting phase transition driven by spin fluctuations, we consider the linearized gap equation

$$\lambda_\mathrm{SC} \Delta^S_{lm}(k) = \frac{T}{N_\mathbf{k}} \sum_{q, l', m'} V^S_{ll', m'm}(q) F_{l'm'}(k - q), \tag{9}$$

for the gap function $\Delta$ with anomalous Green function $F(k) = -G(k) \Delta(k) G^\mathrm{T}(-k)$ in the spin singlet ($S = 0$) or spin triplet pairing channel ($S = 1$) with the respective interactions

$$\hat{V}^{S=0}(q) = \frac{3}{2} \hat{U}^\mathrm{s} \hat{\chi}^\mathrm{s}(q) \hat{U}^\mathrm{s} - \frac{1}{2} \hat{U}^\mathrm{c} \hat{\chi}^\mathrm{c}(q) \hat{U}^\mathrm{c},$$

$$\hat{V}^{S=1}(q) = -\frac{1}{2} \hat{U}^\mathrm{s} \hat{\chi}^\mathrm{s}(q) \hat{U}^\mathrm{s} - \frac{1}{2} \hat{U}^\mathrm{c} \hat{\chi}^\mathrm{c}(q) \hat{U}^\mathrm{c} \tag{10}$$

Constant terms $\sim \hat{U}^\mathrm{s,c}$ were neglected as they did not influence the dominant pairing symmetry. The gap equation represents an eigenvalue problem for $\Delta$ where the eigenvalue $\lambda_\mathrm{SC}$ can be understood as the relative pairing strength of a certain pairing channel. The dominant pairing symmetry of the gap function has the largest eigenvalue $\lambda_\mathrm{SC}$ and the transition temperature is found if $\lambda_\mathrm{SC}$ reaches unity.

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
