# Peer review of "No superconductivity in Pb$_9$Cu$_1$(PO$_4$)$_6$O found in orbital and spin fluctuation exchange calculations"

_SciPost Physics_

## Round 1 · Referee Report · Anonymous (Referee 1) · 2023-10-3

Strengths

1. Solid assessment of the electronic structure of the system.
2. Adequate comparison between FLEX and RPA approaches to the superconducting instability in LK-99.
3. Clear discussion of the pros and cons of the methods.
4. Clear presentation of the results and the overall physics problem under investigation.

Weaknesses

1. Neither FLEX nor RPA ideally suited for strongly correlated systems at strong coupling.
2. Use of approximate crystal data.
3. Doping scenario rather simplified.

Report

This is a sober theory work on the potential of high-Tc superconductivity of LK-99, with and without doping. The system gained a lot of attention this summer, as room-temperature superconductivity at ambient pressure had been claimed.
Here, the authors combine electronic structure calculations with model considerations on the RPA and FLEX level to account for a possible superconducting instability. In more advanced FLEX, no such obvious instability can be found from fluctuating spins or orbitals and thus superconductivity of the claimed kind seemingly excluded.

The performed work is state of the art and the presentation clear and sound. Deficiences in RPA and FLEX to describe strongly correlated electron systems are named, discussed and properly weighed. I support publication as an article in SciPost.

---

## Round 1 · Referee Report · Anonymous (Referee 2) · 2023-10-15

Report

The manuscript attempts to give a theoretical background for electronic properties including superconductivity of LK-99, a material, which was claimed to be superconducting. By now these claims were shown to be incorrect. At the same time a careful theoretical study is anyway useful in order to get a better understanding on potential superconducting compounds. Overall I recommend publication provided the authors give further clarifications:

1) the authors claim that the undoped compound must be a Mott insulator. At the same time the initial electronic structure contain two orbitals per site (d_xz and d_yz) , which in addition to a larger number of nearest neighbors than on a square lattice raises the question on whether this would be always a Mott insulator. What is the reasons for the small bandwidth (nearly flattness) of the xz/yz bands in the DFT, the smallness of the t_pd overlap, weak hopping t_dd hopping. Is there a way to make hoppings larger by applying pressure or uniaxial strain? It would be interesting to discuss.

2) the triangular lattice structure is not uncommon for unconventional superconductors, consider infinite layer cobaltates or organic superconductors. It would be interesting if the authors compare this material with other known unconventional superconductors with traingular lattice structure and discuss the differences and similarities.

3) superconductivity within FLEX has been widely discussed also for the triangular lattices in the context of the systems mentioned above. It woud be nice the authors put some references and stress the differences with those FLEX calculations.

  • validity: good
  • significance: good
  • originality: good
  • clarity: good
  • formatting: good
  • grammar: excellent

Author:  Niklas Witt  on 2023-10-26  [id 4069]

(in reply to Report 2 on 2023-10-15)
Category:
answer to question

We thank the Referee for the careful reading of our manuscript and raising interesting questions. In the following, we want to comment on the points mentioned in the Report:

1) the authors claim that the undoped compound must be a Mott insulator. At the same time the initial electronic structure contain two orbitals per site (d_xz and d_yz) , which in addition to a larger number of nearest neighbors than on a square lattice raises the question on whether this would be always a Mott insulator. What is the reasons for the small bandwidth (nearly flattness) of the xz/yz bands in the DFT, the smallness of the t_pd overlap, weak hopping t_dd hopping. Is there a way to make hoppings larger by applying pressure or uniaxial strain? It would be interesting to discuss.

We extend our gratitude to the referee for posing this insightful question. Since it contains several points, we will address them individually:

  1. Firstly, on the question whether the system would always be a Mott insulator: It is noteworthy that the bandwidth of Cu-$e_ g$ ($d_{yz}+d_{xz}$) is approximately 120 meV, while the interaction $U$ falls within the range of 2-5 eV as indicated by our cRPA calculations and those of others. This substantial ratio of $U/W$~20-40 between $U$ and bandwidth $W$ is sufficiently large to drive the undoped system into a Mott-insulating state. These findings are corroborated by recent publications, including references (Refs. [13-15,28] in the current manuscript, plus Ref. [21] in the revised manuscript):

    Phys. Rev. B 108, L161101 (2023) arXiv:2308.04427 (2023) arXiv:2308.04976 (2023) arXiv:2308.02469 (2023) arXiv:2308.04301 (2023)

    In light of this, we confidently assert that undoped Pb$_9$Cu(PO$_4$)$_6$O exhibits Mott-insulating (or charge-insulating depending on the position of O-$p$ bands) behavior, emphasizing the indispensability of hole or electron doping for the emergence of a metallic state.

  2. Secondly, we want to provide an explanation for the small hopping values between Cu-$e_g$ orbitals. For this, it is essential to highlight that the lattice constants of the relaxed hexagonal crystal structure of LK-99 are $a=b=9.660$ Å and $c=7.226$ Å. Considering the presence of only one Cu ion in each unit cell, the distance between Cu ions is thereby determined to be around 10 Å — nearly two times larger than that of other cuprate superconductors such as SrCuO$_2$ ($a=b=3.93$ Å) and CaCuO$_2$ (a=b=$3.84 $Å). Consequently, the overlap of Wannier functions (spread ~ 8.46 Ų) is small. For LK-99, the hopping between Cu-$d_{yz}$ and $d_{xz}$ is estimated to be in the range of 2-15 meV, while the hopping between Cu-$e_g$ and O-$p$ is in the range of 2-7 meV. Compared to that, the infinite-layer cuprate superconductors SrCuO$_2$ and CaCuO$_2$ have significantly larger hopping between Cu-$d_{x^2-y^2}$ orbitals at -497 meV (SrCuO$_2$) and -522 meV (CaCuO$_2$).

  3. Lastly, in response to the referee's suggestions and comments, we conducted additional DFT structural relaxations, taking external pressure at 0 GPa (the original computations already shown in original manuscript) and new calculations at 5 GPa, i.e., we relaxed the structure and re-compute its band structure and Wannier projections. The computed results include lattice constants, bandwidth and effective hopping between Cu-$e_g$. Notably, external pressure proves effective in reducing lattice constants and enhancing effective hoppings and bandwidth. For instance, applying external pressure of 5 GPa results in reduced lattice constants from $a=9.660$ Šand $c=7.226$ Šat 0 GPa to $a=b=9.464$ Šand $c=7.057$ Šat 5 GPa, and in an increased maximum hopping between $e_g$-1 ($d_{yz}$) and $e_g$-2 ($d_{xz}$) along the (110) direction from 15 meV (0 GPa) to 18 meV (5 GPa), respectively. This can be traced back to the smaller Cu-Cu distance and to more delocalized Wannier functions (spread ~12.03 Ų) since the $d$-$p$ hybridization with neighboring O-$p$ increases. Consequently, the bandwidth of Cu-$e_g$ is increased from ~120 meV to ~180 meV, as shown in the attached Figure.

We add a note at the beginning of Section 2 Results in the revised manuscript. Since the (increased) hopping/bandwidth for the pressurized structure is still very small, we are positive that it has no influence on the results shown and discussed in the paper. Hence, we did not add a detailed discussion which might distract from the main points.

2) the triangular lattice structure is not uncommon for unconventional superconductors, consider infinite layer cobaltates or organic superconductors. It would be interesting if the authors compare this material with other known unconventional superconductors with traingular lattice structure and discuss the differences and similarities.

3) superconductivity within FLEX has been widely discussed also for the triangular lattices in the context of the systems mentioned above. It woud be nice the authors put some references and stress the differences with those FLEX calculations.

Let us respond to these two points in a combined manner: Indeed, many (unconventional) superconductors exist which have a triangular lattice structure, but in most of these systems the origin of superconductivity is not fully understood. Since we discuss the pairing mechanism of spin-(and orbital-)fluctuation-mediated superconductivity in the paper, we focus on this in our comparison (which then, in particular, includes the differences to other FLEX studies). We think that there are two differences to LK-99 which might explain the absence of superconductivity: First, the hopping amplitude in LK-99 is very small (see reply to point 1). This can indicate a small critical temperature $T_{\mathrm{c}}$ because the critical temperature scales with the energy spread of the spin fluctuation excitation in a spin-fluctuation-mediated pairing scenario [see e.g. Scalapino, Rev. Mod. Phys. 84, 1383 (2012) and Fig. 34 therein], similar to $T_{\mathrm{c}}$ scaling with the Debye frequency in phonon-mediated superconductivity. Typically, the energy spread of spin fluctuation excitations scales with the hopping. Hence, the small hopping in LK-99 suggests that superconductivity should have a small $T_{\mathrm{c}}$, if any (well below the 20 K of our calculations). Second, most of the experimentally observed triangular lattice superconductors have a (quasi-)2D electronic structure which was found to be more for spin-fluctuation-based superconductivity than 3D lattice systems such as LK-99 [see Arita et al., J. Phys. Soc. Jpn. 69, 1181 (2000); Monthoux and Lonzarich, Phys. Rev. B 63, 054529 (2001)].

We include a short paragraph in Section 3 Discussion and Conclusion of the revised manuscript where we compare LK-99 to other triangular lattice superconductors, which have been observed experimentally, and related FLEX studies of these systems.

Attachment:

Fig_R1.pdf

---

## Editorial Decision

resubmitted